# CAPSULE NETWORK PROJECTORS ARE EQUIVARIANT AND INVARIANT LEARNERS

## ABSTRACT

Learning invariant representations has been the longstanding approach to self-supervised learning. However, recently progress has been made in preserving equivariant properties in representations, yet do so with highly prescribed architectures. In this work, we propose an invariant-equivariant self-supervised architecture that employs Capsule Networks (CapsNets) which have been shown to capture equivariance with respect to novel viewpoints. We demonstrate that the use of CapsNets in equivariant self-supervised architectures achieves improved downstream performance on equivariant tasks with higher efficiency and fewer network parameters. To accommodate the architectural changes of CapsNets, we introduce a new objective function based on entropy minimisation. This approach which we name CapsIE (**Caps**ule **I**nvariant **E**quivariant Network) achieves state-of-the-art performance across invariant and equivariant tasks on the 3DIEBench dataset compared to prior equivariant SSL methods, while outperforming supervised baselines. Our results demonstrate the ability of CapsNets to learn complex and generalised representations for large-scale, multi-task datasets compared to previous CapsNet benchmarks. *Code is available at redacted*

## 1 INTRODUCTION

Equivariance and invariance have become increasingly important properties and objectives of deep learning in recent times, with precedence being largely placed on the latter. The task of invariance, i.e. being able to classify a specific object no matter the camera perspective or augmentation applied, has driven progress in modern self-supervised learning approaches, specifically those which follow a joint embedding architecture Assran et al. (2022); Bardes et al. (2021); Chen et al. (2020). Equivariance on the other hand is the task of capturing embeddings which equally reflect the translations applied to the input space in the latent space. Equivariance thus has become an important property to capture to enable the learning of high-quality representations in the real world where transformations such as viewpoint are essential.

Self-supervised learning owes its success to invariant objectives, where all recent progress, whether that is by contrastive Chen et al. (2020), information-maximisation Bardes et al. (2021); Zbontar et al. (2021), or clustering based methods Caron et al. (2021); Assran et al. (2022) rely on ensuring invariance in their representations under augmentation. This setting ensures performance in classification based tasks, but when employing the representations in alternative tasks, preservation of information is essential to improve generalisation. To maintain properties of the transformation one can predict the augmentations applied Dangovski et al. (2022); Lee et al. (2021), yet this is typically not considered truly equivariant given the mapping of transformations is not represented in the latent space. Methods that employ such a prediction methodology are typically considered equivariant as the transformation in the input space is preserved in the latent space. Here prediction networks are employed to reconstruct the view prior to transformation Winter et al. (2022), learn symmetric representations Park et al. (2022), or predict the latent representation of the transformed view from the representation of the original view given the transformation parameters Garrido et al. (2023).

The above methods, although promising, enforce equivariance via objective functions on vector representations, yet these methods fail to employ architectural approaches that have shown to be capable of better capturing these properties. Capsule Networks (CapsNets), which utilise a process called routing Sabour et al. (2017); Hinton et al. (2018); Everett et al. (2023); Hahn et al. (2019); De Sousa Ribeiro

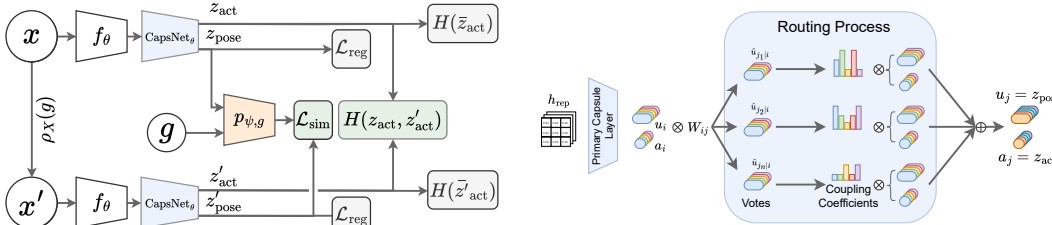

(a) Schematic overview of the CapsIE architecture.    (b) Generalised visualisation of the CapsNet projector.

Figure 1: *Left*: **Schematic overview of the proposed CapsIE architecture.** Representations are fed into a CapsNet projector, and the output embeddings $Z_{\text{act}}$ and $Z_{\text{pose}}$ correspond to invariant and equivariant embeddings respectively. *Right*: **Generalised view of a Capsule projection head.** CNN feature maps are transformed via the primary capsules into poses $u_i$, represented by cylinders, and activations $a_i$, represented by circles. Poses are transformed to votes, which represent a lower-level capsules prediction for each of the higher-level capsules. The routing process then defines how well these votes match the concept represented by the upper-level capsule, creating the coupling coefficients. Coupling coefficients inform $u_j$ and $a_j$, the output of the capsule projector head.

et al. (2020); Liu et al. (2024), are one such architecture, showing signs of desirable properties that other state of the art (SOTA) architectures such as Vision Transformers (ViTs) and CNNs do not. Specifically, CapsNets have shown a natural ability to have strong viewpoint equivariance and viewpoint invariance properties – they achieve this through their ability to capture equivariance with respect to viewpoints in neural activities, and invariance in the weights. In addition, viewpoint changes have nonlinear effects on pixels but linear effects on object relationships De Sousa Ribeiro et al. (2020); Hinton et al. (2018). Ideally, these properties could lead to the development of more sample-efficient models that can exploit robust representations to better generalise to unseen cases and new samples.

However, a common argument is that CapsNets have only shown these properties on toy examples such as the smallNORB dataset LeCun et al. (2004), which many would consider irrelevant for modern architectures. Despite this, small CapsNets outperform much larger CNN and ViT counterparts Everett et al. (2023). In this work, we propose a novel CapsNet formulation and corresponding objective function, achieving SOTA on several experiments and ablations studies on the 3DIEBench dataset Garrido et al. (2023) which has been created to specifically benchmark equivariant and invariant properties of deep learning models. We prove that CapsNets retain their desirable properties on this dataset which is considerably more difficult than what has been previously achieved with CapsNets, while also establishing new SOTA on these tasks.

To summarise, our contributions are:

- We propose a novel architecture based on a Capsule Network projection head that utilises the key assumptions of capsule architectures to learn equivariant and invariant representations which does not require the explicit split of representations.

- We design a new objective function to accommodate the employment of a CapsNet projector, enforcing invariance through entropy minimisation.

- We demonstrate that CapsNet projectors implicitly learn pose understanding in a self-supervised setting.

- We show state of the art performance on 3DIEBench classification for equivariance and invariance benchmark tasks from our CapsNet-based architecture.

## 2 PROBLEM STATEMENT

Typically, self-supervised learning maximises the similarity between embeddings of two augmented views of an image such that they are invariant to augmentations, and instead capture semantically meaningful information of the original image. Views $x$ and $x'$ are each transformed from an image $d \in \mathbb{R}^{c \times h \times w}$ sampled from dataset $\mathcal{D}$ by image augmentations $\tau, \tau' \sim T$ sampled from a set of

augmentations $T$. Embeddings are obtained by feeding each view through an encoder $f_\theta$, where the output representations $y, y'$ are fed through a projection head $h_\phi$ to produce embeddings $z, z'$ whose similarity is maximised. However, it is detrimental in many settings that $f$ is invariant to all transformations, instead in this work we are focused on ensuring that $f$ is equivariant to viewpoint transformations. To train for and evaluate such properties we base our study on the challenging 3DIEBench dataset and corresponding problem definition presented in Garrido et al. (2023).

First, to define equivariance we begin by defining a Group consisting of a set $G$ and a binary operation $\cdot$ on $G$, $\cdot : G \times G \to G$ such that $\cdot$ are associative; there is an identity $e$ which satisfies $e \cdot a = a = a \cdot e, \forall a \in G$; and for each $a \in G$ there exists an inverse $a^{-1}$ such that $a \cdot a^{-1} = e = a^{-1} \cdot a$. Group actions are concerned with how groups manipulate sets, where the left group action can be defined as a function $\alpha$ of group $G$ and set $S$, $\alpha : G \times S \to S$ such that $\alpha(e, s) = s, \forall s \in S$, and $\alpha(g, \alpha(h, s)) = \alpha(gh, s), \forall s \in S$, and $\forall g, h \in G$. In our setting, we are concerned with group representations which are linear group actions acting on vector space $V$, which we define as $\rho : G \to GL(V)$ where $GL(V)$ is the general linear group on $V$. Here $\rho(g)$ describes the transformation applied to both the input data $x$ and latent $f(x)$ given parameters $g$ Park et al. (2022); Garrido et al. (2023). Transformations comprise colour scaling and shifting, and rotations around a fixed point, see Appendix A for further details.

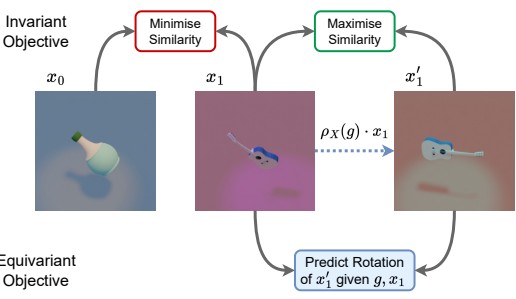

Figure 2: **Visual depiction of the problem statement.** Two images are represented by subscripts $0, 1$ while view under transformation $g$ is given by $'$. Arrows represent the construction of embeddings from an encoder network. ***Top*** the *invariance* objective is to maximise the similarity between embeddings of views originating from the same image. ***Bottom*** the *equivariance* objective aim to learn the transformations $\rho_X(g) \cdot x$ applied to $x$.

Following this, we can define the function $f : X \to Y$ as being equivariant with respect to a group $G$ with representations $\rho_X$ and $\rho_Y$ if $\forall x \in X$, and $\forall g \in G$,

$$f(\rho_X(g) \cdot x) = \rho_Y(g) \cdot f(x). \tag{1}$$

As defined in Garrido et al. (2023) the goal (visually depicted in Figure 2) is to learn $f$ and $\rho_Y$ to construct representations that are equivariant to viewpoint transformations when $\rho_X$ is not known, but the group elements $g$ that parameterise the transformations are known. Further details of these transformations and the benchmark 3DIEBench dataset are given in Section A.

## 3 METHOD

### 3.1 ARCHITECTURE

Our method which we name CapsIE (**Caps**ule Network **I**nvariant **E**quivariant) follows the general joint embedding architecture previously described in Section 2 and extends those proposed by VICReg Bardes et al. (2021) and SIE Garrido et al. (2023). Like previous methods, we employ a ResNet-18 He et al. (2016) encoder as the core feature extractor $f_\theta$ of our network. Yet unlike SIE Garrido et al. (2023), we do not split the representations, and therefore do not require the use of separate invariant and equivariant projection heads. Instead, we employ a single CapsNet (described in Section 3.2) which takes as input the full representation of the encoder in place of the multi-layer perceptron (MLP) in the projection head $h_\phi$. Given the architectural design of CapsNets, our projection head outputs both an activation scalar, representing how active the capsule is, and a $4 \times 4$ pose for each capsule.

To align with the above problem statement and Equation 1, we aim to simultaneously learn invariant and equivariant representations by optimising our network $f$ with respect to the output activations and poses. In this case, we consider the activation vector to capture existence of semantic concepts/objects of the input, thus, the invariant information is preserved by the transformation. The pose on the other hand is designed to encode positional information related to each corresponding capsule Ribeiro et al. (2022) (i.e. semantic concept), therefore it contains equivariant information that that was changed by

the transformation. Akin to the SIE method, we therefore, consider two embedding vectors for each image view, $z_{\text{act}}$ and $z_{\text{pose}}$, which correspond to the capsule activation vector and pose, and invariant and equivariant components respectively.

To enforce our network to learn equivariant properties we utilise a prediction network $p_{\psi,g}$ which takes as input the transformation $g$ and $z_{\text{pose}}$ to predict $z'_{\text{pose}}$, and hence learn $\rho_Y(g)$. In our setting, $g \in \mathbb{R}^3$ corresponding to the quaternions of the rotation applied. In this work, we employ the hypernetwork approach taken by Garrido et al. (2023) which uses an linear projector that takes as input the transformation parameters $g$ to parameterise an MLP predictor. Such a network avoids the case where the transformation parameters $g$ are ignored and the predictor provides invariant solutions. We present more details of the predictor network in the appendix, and visually depict the full CapsIE architecture in Figure 1.

## 3.2 CAPSULE NETWORK PROJECTOR

**Capsule Viewpoint Equivariance** CapsNets are designed to handle spatial hierarchies and recognise objects regardless of their orientation or location, achieving equivariance through their structure Ribeiro et al. (2020). A capsule is a group of neurons – vector-based representations – representing instantiation parameters such as position, orientation, and size. Before any routing process begins, lower-level capsule poses $u_i$ are transformed to $n$ upper-level capsule poses $u_{j|i}$ which align with concepts represented by higher-level capsules, preserving spatial relationships and hierarchical information. It is then determined through the routing process how well these transformed poses correspond with the concept represented by the upper-level capsule. CapsNets, unlike convolution, excel in achieving viewpoint invariance and viewpoint equivariance as they can capture equivariance with respect to viewpoints in neural activities, and invariance in the network's weights Ribeiro et al. (2022). Consequently, capsule routing aims to detect objects by looking for agreement between their parts, thereby performing equivariant inference.

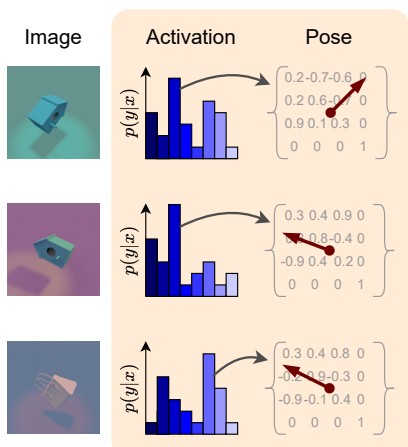

Figure 3: **Simplified visual representation of CapsNet outputs.** Activation vector outputs the probability of each capsule being activated, whereas the pose matrix corresponds to the object pose in relation to the frame.

**Self Routing Capsule** We use the Self Routing CapsNet (SRCaps) Hahn et al. (2019) based on the efficiency of its non-iterative routing algorithm. We consider the trade-off of a small amount of classification accuracy to be acceptable when comparing the performance of SRCaps to other capsule architectures which require significantly more resources to train. Based on the size of the 3DIEBench dataset, these other routing algorithms would be unsuitable.

SRCaps calculates the coupling coefficients between each capsule in lower layer $i$ with each capsule in upper layer $j$ to produce the coupling coeffients $c_{ij}$. It does so by using a learnable routing matrix $W^{route}$ multiplied with the lower capsule pose vector $u_i$, mimicking a single layer perceptron to produce routing coefficients $b_{ij}$ which when passed through a softmax function produce coupling coefficients $c_{ij}$. Additionally, we determine the activation of upper-level capsules $a_j$ by first multiplying $a_i$ by $c_{ij}$ to create votes and then dividing this by $a_i$ to create weighted votes.

$$c_{ij} = \text{softmax}(W_i^{route} u_i)_j, \quad a_j = \frac{\sum_{i \in \Omega_l} c_{ij} a_i}{\sum_{i \in \Omega_l} a_i} \quad (2)$$

The output pose of a capsule layer is calculated using learnable weight matrix $W^{pose}$ which when multiplied with $u_i$ provides a capsule pose of each lower-level capsule for each upper-layer capsule i.e $u_{j|i}$. Following the same procedure as the activations, $u_j$ is the weighted sum of these poses by $a_j$.

$$\hat{u}_{j|i} = W_{ij}^{\text{pose}} u_i, \quad u_j = \frac{\sum_{i \in \Omega_l} c_{ij} a_i \hat{u}_{j|i}}{\sum_{i \in \Omega_l} c_{ij} a_i}. \quad (3)$$

### 3.3 OBJECTIVE FUNCTIONS

**Invariant Criterion.** To train our aforementioned architecture we first introduce an invariant objective as the cross entropy between activation probability vectors, $H(Z_{\text{act}}, Z'_{\text{act}})$ where $Z$ refers to the matrix embeddings over a batch. The aims is to enforce embedding probability pairs originating from the same image to be matched. To avoid trivial solutions and collapse to a single capsule, we employ the mean entropy maximisation regularisation Assran et al. (2021; 2022) on the same activation probability vectors to encourage the model to utilise the full set of capsules over a batch. This regularisation maximises the entropy of the mean probabilities $H(\bar{Z}_{\text{act}})$ and $H(\bar{Z}'_{\text{act}})$, where $\bar{Z}_{\text{act}} = \frac{1}{B} \sum_{i=1}^{B} Z_{\text{act}}$ and $B$ is the batch size.

**Equivariant Criterion.** As previously stated in Section 2, our goal is to learn the predictor $p_{\psi,g}$ to model $\rho_Y(g)$ as to enforce equivariant representations. This is achieved by maximising the cosine similarity between the output vector of the predictor $p_{\psi,g}(Z_{\text{pose}})$ given translation parameters $g$ and equivariant representation $Z_{\text{pose}}$, and the augmented view's equivariant representation vector $Z'_{\text{pose}}$. To avoid collapse and improve training stability we also regularise the output of $p_{\psi,g}(Z_{\text{pose}})$ by ensuring the variance of the predicted equivariance representation is 1 to avoid collapse. Whereas, SIE Garrido et al. (2023) finds this to be an optional but recommended component, we found in practice, without such regularisation the predictor would consistently collapse to trivial solutions.

As with the activation vector we employ variance-covariance regularisation on the pose to ensure they do not collapse of representations to trivial solutions. The variance objective $V$ ensures that all dimensions $d$ in the embedding vector are equally utilised while the covariance objective $C$ decorrelates the dimensions to reduce redundancy across dimensions. The regularisation for equivariant vectors $\mathcal{L}_{\text{reg}}$ is given by

$$\mathcal{L}_{\text{reg}}(Z) = \lambda_C\ C(Z) + \lambda_V\ V(Z), \quad \text{where} \tag{4}$$

$$C(Z) = \frac{1}{d} \sum_{i \neq j} Cov(Z)_{i,j}^2 \quad \text{and} \quad V(Z) = \frac{1}{d} \sum_{j=1}^{d} \max\left(0, 1 - \sqrt{Var(Z_{\cdot,j})}\right). \tag{5}$$

The final objective function is given by the weighted sum of the individual objectives

$$\mathcal{L}(Z_{\text{act}}, Z'_{\text{act}}, Z_{\text{pose}}, Z'_{\text{pose}}) = \lambda_{\text{inv}} H(Z_{\text{act}}, Z'_{\text{act}}) + (H(\bar{Z}_{\text{act}}) + H(\bar{Z}'_{\text{act}})) + \tag{6}$$

$$\lambda_{\text{equi}} \frac{1}{N} \sum_{i=1}^{N} \|p_{\psi,g_i}(Z_{i,\text{pose}}) - Z'_{i,\text{pose}}\|_2^2 + \tag{7}$$

$$\mathcal{L}_{\text{reg}}(Z_{\text{pose}}) + \mathcal{L}_{\text{reg}}(Z'_{\text{pose}}) + \lambda_V V(p_{\psi,g_i}(Z_{i,\text{pose}})). \tag{8}$$

## 4 EXPERIMENTATION

### 4.1 TRAINING PROTOCOL

To directly compare with prior works employing the 3DIEBench dataset, we follow an identical training protocol, as defined in Garrido et al. (2023). All methods employ a ResNet-18 encoder network ($f_\theta$), for the projection head ($h_\phi$) we compare a variety of hyperparameterisations, which we later describe in the following sections. For primary bench-marking we train our model for 2000 epochs using the Adam Kingma & Ba (2014) optimiser with default settings, a fixed learning rate of 1e-3 and batch size of 1024. For ablations and sensitivity analyses we train for 500 epochs and employ a batch size of 512, with other settings remaining unchanged. We find in practice that 500 epochs presents a strong correlation of performance. By default the objective function weighting are as follows, $\lambda_{\text{inv}} = 0.1$, $\lambda_{\text{equi}} = 5$, $\lambda_V = 10$, $\lambda_C = 1$. Each self-supervised 2000 epoch pretraining run took approximately 22 hours using 3 Nvidia A100 80GB GPU's, with 64 capsule models taking approximately 25 hours using 6 Nvidia A100 80GB GPU's. All eval tasks are completed on a single Nvidia A100 80GB GPU and take approximately 6 hours for angle prediction, and 3 hours for classification.

Table 1: **Evaluation of invariant properties via downstream classification task.** Representations are learnt under the invariance and rotation equivariant objective, we evaluate both the representations and the intermediate embeddings of the projection head under varying number of capsules. FLOPs and Parameters correspond to computation during training, *'-' refers to non-compatible experiments*.

| Method | Computational Load | | Embedding Dims | | Classification (Top-1%) | | |
| | Parameters | # FLOPs | Inv. | Equi. | All | Inv. | Equi. |
|---|---|---|---|---|---|---|---|
| *Supervised* | | | | | | | |
| ResNet-18 | 11.2M | 3.09G | - | - | 87.47 | - | - |
| SR-Caps - 16 | 11.0M | 3.16G | - | - | - | 73.85 | - |
| SR-Caps - 32 | 13.0M | 4.27G | - | - | - | 59.70 | - |
| SR-Caps - 64 | 18.7M | 8.22G | - | - | - | 69.45 | - |
| *Encoder Representation* | | | | | | | |
| SIE | 20.1M | 13.07G | 512 | 512 | 82.94 | 82.08 | 80.32 |
| CapsIE - 16 | 12.7M | 3.49G | 16 | 256 | 78.96 | - | - |
| CapsIE - 32 | 14.7M | 4.57G | 32 | 512 | 80.00 | - | - |
| CapsIE - 64 | 20.4M | 8.69G | 64 | 1024 | 80.26 | - | - |
| *Projector - 1st Intermediate Embedding* | | | | | | | |
| SIE | 20.1M | 13.07G | 512 | 512 | - | 80.53 | 77.64 |
| CapsIE - 16 | 12.7M | 3.49G | 16 | 256 | - | 82.96 | - |
| CapsIE - 32 | 14.7M | 4.57G | 32 | 512 | - | 83.49 | - |
| CapsIE - 64 | 20.4M | 8.69G | 64 | 1024 | - | **83.64** | - |

## 4.2 DOWNSTREAM EVALUATION

To evaluate the quality of representations learnt under self-supervision, we use the standard benchmark approach of learning downstream task specific networks with frozen representations as input. In our case we evaluate the representations in three distinct tasks to evaluate both invariant and equivariant properties, we use a linear evaluation training protocol of the frozen representations. Further details of the evaluation protocol are given in the appendix B.2.

**Invariant Evaluation.** To evaluate invariant properties of the representation we train a classifier on either the frozen representations output from the encoder network or the intermediate embeddings of the capsule network projector. Given our advocation for CapsNets we evaluate using both the standard linear classification and a capsule layer whose number of out capsules is set to the number of classes. All methods are trained for 300 epochs by cross entropy.

**Equivariant Evaluation.** Evaluating equivariant properties is achieved through a rotation prediction task in which a three layer MLP is trained to predict the quaternions defining the rotation between two views of the same object. We train for 300 epochs using MSE loss. Similarly to rotation prediction, we evaluate the representation's equivariant properties by regressing the the colour hue of an object view. We train a single linear layer for 50 epochs using MSE loss.

**Representation Quality.** The performance of CapsIE for both invariant and equivariant benchmark tasks is given in Tables 1 and 2 respectively. We evaluate both the representations produced by the ResNet-18 encoder and the intermediate embeddings of the capsule layer projection head given different values for the numbers of capsules. We observe that across all models that the invariant properties captured within the representations marginally suffer compared to the MLP projector of SIE. This observation is expected given the significantly reduced number of embeddings employed in the invariant criterion compared to SIE. However, the evaluation of equivariant properties captured by the representations demonstrates that the use of a capsule projector in place of a MLP can lead to vastly improved performance in rotation prediction ($\uparrow 0.05\ R^2$) advancing the prior state-of-the-art whist also improving on the supervised baseline by a significant margin.

Additionally, we observe that the colour prediction task achieves a performance close to that of the supervised setting even though our criteria do not directly optimise for such equivariant properties. This suggests that the CapsNet is responsible for implicitly capturing transformations of the input whilst having little to no impact on the tasks directly optimised for.

Table 2: **Evaluation of equivariant properties via downstream rotation prediction (*left*) and colour prediction (*right*) tasks.** Representations are learnt under the invariance and rotation equivariant objective, we evaluate both the representations and the intermediate embeddings of the projection head under varying number of capsules. *'-' refers to non-compatible experiments*.

| | Rotation Prediction ($R^2$) | | | Colour Prediction ($R^2$) | | |
|---|---|---|---|---|---|---|
| Method | All | Inv. | Equi. | All | Inv. | Equi. |
| *Supervised* | | | | | | |
| ResNet-18 | 0.76 | - | - | 0.99 | - | - |
| SR-Caps - 16 | - | - | 0.83 | - | - | 0.99 |
| SR-Caps - 32 | - | - | 0.84 | - | - | 0.99 |
| SR-Caps - 64 | - | - | 0.80 | - | - | 0.99 |
| *Encoder Representation* | | | | | | |
| SIE | 0.73 | 0.23 | 0.73 | 0.07 | 0.05 | 0.02 |
| CapsIE - 16 | **0.78** | - | - | **0.97** | - | - |
| CapsIE - 32 | 0.75 | - | - | **0.97** | - | - |
| CapsIE - 64 | 0.72 | - | - | **0.97** | - | - |
| *Projector - 1st Intermediate Embedding* | | | | | | |
| SIE | - | 0.38 | 0.58 | - | 0.45 | 0.09 |
| CapsIE - 16 | - | - | **0.78** | - | - | **0.97** |
| CapsIE - 32 | - | - | 0.77 | - | - | **0.97** |
| CapsIE - 64 | - | - | **0.78** | - | - | **0.97** |

**Intermediate Projector Embeddings.** The role of the projector is primarily employed to decorrelate the embeddings on which the objective function operates on from the representations employed downstream. The premise is to avoid representations that are over-fit to the self-supervised objective Bordes et al. (2022). However, it has been well studied that it can be beneficial to maintain a number of projector layers and instead utilise intermediate projector embeddings for downstream tasks. In our case, the preservation of capsule layers for downstream tasks ensures that the desirable equivariant and part-whole properties are maintained. Specifically, the equivariant information that has shown to be captured in the object pose Ribeiro et al. (2022).

We evaluate the intermediate embeddings output from the primary capsule layer in the same manner as the representations, however the activations and pose are given over a spatial region we perform average pooling to return an activation vector and a $4 \times 4$ pose for each capsule which we then flatten into a vector. As with the representation evaluation we report our invariant and equivariant task performance in Tables 1 and 2 respectively. We find across all settings that evaluating the intermediate embeddings of the capsule projector leads to improved performance on all tasks. We observe that classification via the activation vector (i.e. invariant part) significantly improves on SIE ($\uparrow$ 0.7 Top-1%) while approaching performance levels of explicitly invariant approaches such as VICReg (see appendix C.2). The same increase in performance is seen for the rotation prediction task, where evaluating on the pose of the intermediate embedding (equivariant part) leads to improved performance across all capsule based models, extending beyond supervised training.

## 4.3 QUANTITATIVE EVALUATION OF EQUIVARIANCE

In order to quantitatively evaluate the equivariant performance of our method and capsule projector, we employ commonly used metrics including Mean Reciprocal Rank (MRR) and Hit Rate at k (H@k). We utilise the same setup as described in Garrido et al. (2023), and discuss in further details the setup in appendix B.3. All the results evaluated by the aforementioned metrics are given in Table 3. Our CapsIE network outperforms EquiMod, Only Equivariance and SIE by a considerable margin across all metrics and for all dataset splits. We achieve strong perfromance on PRE, reporting 0.21 PRE on the validation set compared to 0.48 for EquiMod and Only Equivariance and 0.29 for SIE. The same large gains in equivariant performance are shown for MRR and H@1 and H@5. Note, a random H@1 results in a performance for 2% (0.02) demonstrating that our method lies well above random.

Table 3: **Quantitative evaluation of the predictor when using a Capsule network projector, using PRE, MRR and H@k.** The source dataset which embeddings are computed and the dataset used for retrieval are given in the format *source-retrieval* for PRE and *source* for MRR and H@k.

| | PRE ($\downarrow$) | | | MRR ($\uparrow$) | | H@1 ($\uparrow$) | | H@5 ($\uparrow$) | |
| Method | train-train | val-val | val-all | train | val | train | val | train | val |
|---|---|---|---|---|---|---|---|---|---|
| EquiMod | 0.47 | 0.48 | 0.48 | 0.17 | 0.16 | 0.06 | 0.05 | 0.24 | 0.22 |
| Only Equivariance | 0.47 | 0.48 | 0.48 | 0.17 | 0.17 | 0.06 | 0.05 | 0.24 | 0.22 |
| SIE | 0.26 | 0.29 | 0.27 | 0.51 | 0.41 | 0.41 | 0.30 | 0.60 | 0.51 |
| CapsIE | **0.17** | **0.21** | **0.20** | **0.60** | **0.47** | **0.50** | **0.36** | **0.71** | **0.58** |

## 5 ABLATIONS & SENSITIVITY ANALYSES

### 5.1 NUMBER OF CAPSULES

Each capsule, in theory, should represent a unique concept, thus when the number of capsules is increased, logically so should the networks representation ability to capture an increasing number of semantic concepts. Observing the invariant performance during training (Figure 4) and downstream evaluation in Table 1, CapsIE gains slight improvement with the addition of more capsules. This demonstrates that our model has better utilised the additional representational power to improve performance. This pattern is also observed when evaluating equivariant properties (shown in Figure 4), yet is less pronounced. However, in Table 1 we also show that increasing the number of capsules of a supervised SR-Caps model trained in a standard supervised fashion is not an indicator of increased performance, aligning with prior capsule research Everett et al. (2023). This differentiation in behaviour provides an interesting direction for future research.

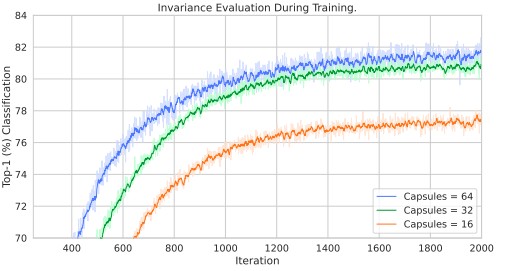 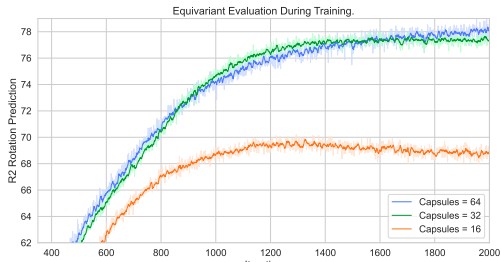

Figure 4: **Invariant and equivariant performance during training when varying number of capsules.** *(left)* Classification evaluation performance (top-1 %). *(right)* Rotation prediction evaluation performance ($R^2$).

### 5.2 IMPLICIT VIEWPOINT EQUIVARIANCE

To investigate whether capsule models are implicitly learning equivariant properties without any explicit enforcing criterion, we train our CapsIE model to predict colour hue rather than rotation via the predictor network. Given the observation in Table 2 that CapsIE is able to achieve near-perfect prediction without any optimising criterion, we hypothesise that capsules are implicitly capable of learning equivariant properties. The evaluation performance of representations on the rotation task under the aforementioned ablation pre-training settings in Table 4, demonstrates that CapsNets indeed learns more implicit equivariant properties than the benchmark SIE by a considerable margin. We do however observe that SIE is still able to capture some equivariant information without being explicitly trained for,

Table 4: Evaluation of SIE and CapsIE - 32 downstream rotation prediction on representations by either a rotation or colour hue equivariant objective.

| Rotation Prediction ($R^2$) | |
|---|---|
| Method | All |
| SIE - Rotation | 0.43 |
| SIE - Colour | 0.29 ($\downarrow$ 0.14) |
| CapsIE - Rotation | 0.59 |
| CapsIE - Colour | 0.48 ($\downarrow$ 0.11) |

which does not align with the findings of the setting where colour is evaluated Garrido et al. (2023). Notably, this results is not a significant improvement over random representations in which the $R^2$ lies approximately at 0.25. This observation presents an interesting study for future work, while the improvement of the CapsIE model over SIE empirically demonstrates the viewpoint equivariant assumptions of CapsNets.

# 6 RELATED WORK

## 6.1 EQUIVARIANT SELF-SUPERVISED LEARNING

Self-supervised learning has seen the majority of its success in the invariant setting by either contrastive Chen et al. (2020), information maximisation Zbontar et al. (2021); Bardes et al. (2021), or clustering methods Caron et al. (2021); Assran et al. (2022). All families of approaches rely on training a network to be invariant to transformations by increasing the similarity between embeddings of the same image under augmentation. The differing approaches emerge from alternative methods to avoid collapse, the phenomena where embeddings fall into a lower-dimensional subspace rather than the entire available embedding space resulting in a trivial solution Hua et al. (2021). Although these methods differ, they all produce similarly performing representations, hence we employ information maximisation methods as the basis of this work due to their computational efficiency.

Learning to be invariant to transformations is typically useful for semantic discrimination tasks, yet preserving information about the transformations can be highly beneficial. Some approaches have attempted to capture specific information regarding transformations by predicting the applied augmentation parameters Lee et al. (2021), preserving the strength of augmentations Xie et al. (2022) and introducing rotational transformations Dangovski et al. (2022). However, as stated in Garrido et al. (2023), these methods provide no guarantee that a mapping is learnt in the latent space that reflects the transformations in the input space. Hence, methods have been employed that address this limitation, Devillers & Lefort (2022); Park et al. (2022); Garrido et al. (2023) all employ predictor networks to predict the displacement representations in the latent space given one view representation and the transformation parameters. The latter, SIE Garrido et al. (2023), is the basis of our work, which further extends prior methods by splitting representation vectors into invariant and equivariant parts to better separate differing information.

## 6.2 CAPSULE NETWORKS

CapsNets present an alternative architecture to CNNs, addressing their limitations by explicitly preserving hierarchical spatial relationships between features Sabour et al. (2017). CapsNets replace scalar neurons with vector or matrix poses, representing specific concepts at different levels of a parse tree as the network goes deeper. The first layer (primary capsules) corresponds to the most basic parts, while capsules in deeper layers represent more complex concepts made up of the simpler concepts as they get closer to the final layer where each capsule corresponds to a specific class.

The key components of the CapsNet are the pose and the activation. The pose of a capsule is an embedding vector or matrix which provides a representation for the concept. The activation scalar is a value between 0 and 1 which represents how certain the network is that the concept is present and can be calculated directly from the values of the pose or via other means via the routing mechanism.

The key novelty in CapsNets is the routing mechanism, which determines the contributions of lower-level capsules to higher-level capsules. Numerous routing algorithms, both iterative and non-iterative, have been proposed to address the efficiency and effectiveness of this process Sabour et al. (2017); Mazzia et al. (2021); Feng et al. (2024); Ribeiro et al. (2020); Hinton et al. (2018); De Sousa Ribeiro et al. (2020); Yang et al. (2021); Everett et al. (2024). Among these, SRCaps Hahn et al. (2019) introduces a non-iterative routing mechanism. This method maintains all of the desirable properties of CapsNets, such as equivariance while largely mitigating the time cost of iterative methods, at the price of a small amount of performance. However, SRCaps faces the same limitations as other CapsNets where high resolution or high class datasets are beyond the network's abilities when trained in a standard fashion. For a more detailed description of capsule routing mechanisms please check this review Ribeiro et al. (2022).

# 7 CONCLUSION

Our proposed method demonstrates how self-supervised CapsNets can be employed to better learn equivariant representations, leveraging architectural assumptions removing the the need to explicitly split representation vectors and train separate projector networks. The resulting solution, CapsIE, achieves state-of-the-art performance in equivariant and invariant downstream benchmarks with a significant improvement of 0.05 $R^2$ on prior self-supervised rotation prediction tasks and 0.02 $R^2$ improvement over the supervised baseline. In addition, we observe competitive performance of CapsIE on equivariant tasks not explicitly trained for, further demonstrating the implicit equivariant properties of our capsule architecture under standard invariant optimisation criteria akin to those of VICReg Bardes et al. (2021), SimCLR Chen et al. (2020), and MSN Assran et al. (2022). Our results contribute significantly to the avocation of CapsNets in self-supervised representation learning, introducing desirable properties with improved effectiveness over MLP projectors.

**Ethics Statement.** This work aims to learn higher quality and more applicable representations of images without human generated annotations, therefore such methods can lead to positive societal impacts the development of more accurate or informative models for a number of downstream tasks. However, as is the case with all vision systems, there is potential for exploitation and security concerns and one should take into consideration AI misuse when extending our method.

In addition, the use of CapsNets has shown little improvement when scaled, similar to the findings of Everett et al. (2023); Mitterreiter et al. (2023); Nair et al. (2021) where the addition of more capsule layers has either led to stagnated or decreased performance. We strongly feel that our work should be revisited when a capsule network that can both scale and retain the desirable properties of capsule networks is found.

As stated in the original dataset proposal and the problem setting the methodology presented rely on the group elements being known. Hence, the applicability of the proposed method is only possible in settings where group elements are known. However, we present findings where additional group elements are preserved without prior knowledge of the group, this suggests the CapsNets are more capable than previous invariant methods at capturing equivariant properties, thus opening an intriguing direction for future work.

**Reproducibility Statement.** We have included the source code required to reproduce all experiments and ablation studies presented in the main body of the paper and the appendix. We also provide the details of our training protocol in the appendix.

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

## A  3DIEBench Dataset

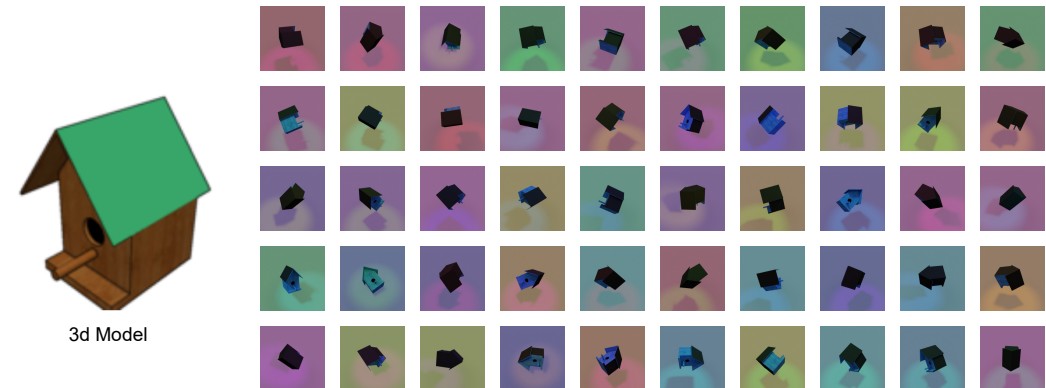

Figure 5: **The 3DIEBench dataset**, one 3d model is used to create 50 different views in a synthetic environment, which are saved as images along with the latent values by which they are transformed.

Typical equivariant datasets are generally handcrafted and simple, with a small amount of classes and instances within each class. This is due to the time needed in order to ensure correctness. While standard image datasets do allow for testing invariance in the form of augmenting the same image in two different ways, they do not allow for precise transformation of the subject. Thus the need for a new, synthetic dataset.

We use the 3DIEBench Garrido et al. (2023) dataset[1], which has been created specifically to be a hard yet controlled test-bed for invariant and equivariant methods. The dataset consists of 52,472 3d objects across 55 classes of 3d objects from ShapeNetCorev2 Chang et al. (2015) posed in 50 different views as well as the latent information of the view, this can be seen in figure 5. For training we then randomly select two views from each model in the training set. The parameters by which the model could have been augmented can be seen in table 5.

Table 5: **Values of the factors of variation used for the generation of 3DIEBench.** Each value is sampled uniformly from the given interval. Object rotation is generated as Tayt-Bryan angles using extrinsic rotations. Light position is expressed in spherical coordinates. This table is sourced from Garrido et al. (2023).

| Parameter | Minimum value | Maximum value |
|---|---|---|
| Object rotation X | $-\frac{\pi}{2}$ | $\frac{\pi}{2}$ |
| Object rotation Y | $-\frac{\pi}{2}$ | $\frac{\pi}{2}$ |
| Object rotation Z | $-\frac{\pi}{2}$ | $\frac{\pi}{2}$ |
| Floor hue | 0 | 1 |
| Light hue | 0 | 1 |
| Light $\theta$ | 0 | $\frac{\pi}{4}$ |
| Light $\phi$ | 0 | $2\pi$ |

[1]The full dataset and splits employed can be found at `https://github.com/facebookresearch/SIE`

## B  TRAINING PROTOCOLS

### B.1  CAPSIE PRE-TRAINING

Our proposed CapIE model is comprised of a ResNet-18 encoder, SR-CapsNet comprised of a primary capsule layer routed to a second capsule layer. The SR-CapsNet projector takes as input the activation map output of the ResNet prior to the final global average pooling. The predictor network employed is that described in Garrido et al. (2023). All details of the architectural design are given in the main paper.

Training of our CapsIE model is done over 2000 epochs with a batch size of 1024, optimised via the Adam optimiser with learning rate 0.001, and default parameters, $\beta_1 = 0.9, \beta_2 = 0.999$. By default the objective function weighting are as follows, $\lambda_{\text{inv}} = 0.1$, $\lambda_{\text{equi}} = 5$, $\lambda_V = 10$, $\lambda_C = 1$ where we empirically found these optimal for our setting. Further performance gains could be achieved by the tuning of such parameters, however, we deemed this unnecessary.

For ablation studies, where we explicitly state, we train for fewer epochs and with a smaller batch size, 500 and 512 respectively. We find in practice this setting is a strong proxy for full training performance and significantly save computational resource.

Training time for 2000 epochs with batch size of 1024, as previously stated, took approximately 22 hours using 3 Nvidia A100 80GB GPU's, with 64 capsule models taking approximately 25 hours using 6 Nvidia A100 80GB GPU's.

### B.2  DOWNSTREAM EVALUATION

In our work we perform evaluation on both the frozen ResNet-18 representations and the representations from the primary capsules layer, which are evaluated using either a linear classifier or additional capsule layer acting as a class capsules, i.e the number of capsules is set to the number of classes and activations are used as the logits. Here, we detail the exact training protocols to ensure complete reproducability.

For our evaluations we use two different depths of MLP heads, these are: 1. **Deep MLP** referring to an MLP with layers containing in_dim - 1024 - out_dim neurons, with intermediate ReLU activations. 2. **Shallow MLP** referring to a single MLP layer with in_dim in neurons and out_dim out neurons. When we evaluate our primary capsules for angle and colour prediction, we average the 8x8 feature map so that we only have a single pose vector for the entire image. For our Capsule Classification task, we do not have an in_dim as we do not use a MLP, but instead use a capsule layer which operates on the primary capsules pose and activations.

Table 6: **Training settings for our evaluations.** Settings are the same for all number of capsules. **NC** is used as shorthand for number of capsules. - denotes that this element is not used. * denotes multiplication.

|  | Representations Angle | Representations Colour | Representations Classification | Capsule Angle | Capsule Colour | Capsule Classification |
|---|---|---|---|---|---|---|
| Caps Head | - | - | - | - | - | Yes |
| MLP Head | Deep | Shallow | - | Deep | Shallow | - |
| in_dim | 512 | 512 | 512 | **NC** * 16 | **NC** * 16 | N/A |
| out_dim | 4 | 2 | 55 | 4 | 2 | 55 |
| Optimizer | Adam | Adam | Adam | Adam | Adam | Adam |
| LR | 0.001 | 0.001 | 0.001 | 0.001 | 0.001 | 0.001 |
| $\beta_1$ | 0.9 | 0.9 | 0.9 | 0.9 | 0.9 | 0.9 |
| $\beta_2$ | 0.999 | 0.999 | 0.999 | 0.999 | 0.999 | 0.999 |
| Batch Size | 256 | 256 | 64 | 256 | 256 | 256 |
| Epochs | 300 | 50 | 300 | 300 | 50 | 300 |
| Objective | MSE | MSE | Cross Entropy | MSE | MSE | Cross Entropy |

### B.3 QUANTITATIVE EQUIVARIANT EVALUATION IMPLEMENTATION DETAILS

We evaluate the equivariant properties of the predictor in line with that proposed in Garrido et al. (2023) reporting the Mean Reciprocal Rank (MRR) and Hit Rate at k (H@k) on the multi-object setting. Given a source and target pose of an object, we first compute the embeddings of each image, and pass the source embedding through the predictor and use the resulting vector to retrieve the nearest neighbours.

The MRR is the average reciprocal rank of the target embedding in the retrieved nearest neighbour graph. H@k in this case is computed to be 1 if the target embedding is in the k-NN graph of the predicted embedding, where we only look for nearest neighbours among the views of the same object.

The Prediction Retrieval Error (PRE) gives an evaluation of predictor quality, and is given by the distance between its rotation $q_1 \in \mathbb{H}$ and the target rotation $q_2$ as $d = 1 - <q_1, q_2>^2$ of the nearest neighbour of the predicted embedding averaged over the whole dataset. Full implementation details can be found in the open-source code provided.

### B.4 SUPERVISED TRAINING OF SR-CAPS

In our work we train a Self Routing Capsule Network model in a supervised fashion for the downstream tasks to evaluate whether our pretrained model improves the quality of downstream evaluations. The training setting of these runs can be found in table 7.

Deep MLP refers to an MLP with layers containing number_caps * 16 * 2 - 1024 - 4 neurons, with intermediate ReLU activations. Shallow MLP head refers to a single MLP layer with number_caps * 16 * 2 in neurons and either 4 (for rotation prediction) or 2 (for colour prediction) out neurons.

Table 7: **Training settings for our supervised Self Routing Capsule Network model.** Settings are the same for all number of capsules. - denotes that this element is not used.

|  | Angle | Colour | Classification |
|---|---|---|---|
| Caps Head | - | - | Yes |
| MLP Head | Deep | Shallow | - |
| Optimizer | Adam | Adam | Adam |
| LR | 0.001 | 0.001 | 0.001 |
| $\beta_1$ | 0.9 | 0.9 | 0.9 |
| $\beta_2$ | 0.999 | 0.999 | 0.999 |
| Batch Size | 256 | 256 | 64 |
| Epochs | 300 | 50 | 300 |
| Objective | MSE | MSE | Cross Entropy |

Table 8: **Capsule downstream invariance evaluation on projection head embeddings.**

|  | Embedding Dims | | Classification (Top-1%) | |
| --- | --- | --- | --- | --- |
| Method | Inv. | Equi. | Representations | Embeddings |
| Supervised | - | - | 87.47 | |
| SIE | 512 | 512 | 82.94 | |
| *Capsule Projector Naive Evaluation* | | | | |
| CapsIE - 16 | 16 | 256 | 78.96 | 65.83 |
| CapsIE - 32 | 32 | 512 | 80.00 | 69.12 |
| CapsIE - 64 | 64 | 1024 | 80.26 | 56.64 |

Table 9: **Evaluation of invariant properties on downstream classification task for baseline SSL methods.** We evaluate both the representations and the intermediate embeddings of the projection head when different numbers of capsules in the projection head is used. *'-' refers to non-compatible experiments*.

|  | Embedding Dims | | Classification (Top-1%) | | |
| --- | --- | --- | --- | --- | --- |
| Method | Inv. | Equi. | All | Inv. | Equi. |
| *Encoder Representation* | | | | | |
| VICReg | - | - | 84.74 | - | - |
| VICReg, $g$ kept identical | - | - | 72.81 | - | - |
| SimCLR | - | - | 86.73 | - | - |
| SimCLR, $g$ kept identical | - | - | 71.21 | - | - |
| SimCLR + AugSelf | - | - | 85.11 | - | - |
| EquiMod (Original predictor) | - | - | **87.19** | - | - |
| EquiMod (SIE predictor) | - | - | **87.19** | - | - |
| SIE Garrido et al. (2023) | 512 | 512 | 82.94 | 82.08 | 80.32 |
| SIE * | 512 | 512 | 82.54 | 82.11 | 80.74 |
| CapsIE - 16 | 16 | 256 | 78.96 | - | - |
| CapsIE - 32 | 32 | 512 | 80.00 | - | - |
| CapsIE - 64 | 64 | 1024 | 80.26 | - | - |
| *Capsule Projector - 1st Intermediate Embedding* | | | | | |
| CapsIE - 16 | 16 | 256 | - | 82.96 | - |
| CapsIE - 32 | 32 | 512 | - | 83.49 | - |
| CapsIE - 64 | 64 | 1024 | - | **83.64** | - |

## C  FURTHER EXPERIMENTATION

### C.1  CAPSULE DOWNSTREAM CLASSIFICATION ON EMBEDDINGS.

Table 8 provides the downstream classification evaluation for the frozen output embeddings of the CapsNet projection head. The drop in performance is an expected result inline with that demonstrated by Chen et al. (2020); Bordes et al. (2022). These results signify the importance of the projection head and specifically its role in decorrelating the embeddings from representations to avoid overfitting to the pre-training objective. We do note however, that our observed performance drop is inline or slightly less than that reported in Garrido et al. (2023).

### C.2  INVARIANT AND EQUIVARIANT SSL BENCHMARKS

We report below the classification (invariant, Table 9), rotation prediction, and colour prediction (equivariant, Table 10) performance of baseline self-supervised methods. The below baseline results are acquired from Garrido et al. (2023), with the exception of those denoted by '*' which corresponds to our re-implementation.

Table 10: **Evaluation of equivariant properties on downstream rotation prediction (*left*) and colour prediction (*right*) tasks for baseline SSL methods.** We evaluate both the representations and the intermediate embeddings of the projection head when different numbers of capsules in the projection head is used.

| Method | Rotation Prediction ($R^2$) | | | Colour Prediction ($R^2$) | | |
|---|---|---|---|---|---|---|
| | All | Inv. | Equi. | All | Inv. | Equi. |
| Encoder Representation | | | | | | |
| VICReg | 0.41 | - | - | 0.06 | - | - |
| VICReg, $g$ kept identical | 0.56 | - | - | 0.25 | - | - |
| SimCLR | 0.50 | - | - | 0.30 | - | - |
| SimCLR, $g$ kept identical | 0.54 | - | - | 0.83 | - | - |
| SimCLR + AugSelf | **0.75** | - | - | 0.12 | - | - |
| EquiMod (Original predictor) | 0.47 | - | - | 0.21 | - | - |
| EquiMod (SIE predictor) | 0.60 | - | - | 0.13 | - | - |
| SIE Garrido et al. (2023) | 0.73 | 0.23 | 0.73 | 0.07 | 0.05 | 0.02 |
| SIE * | 0.72 | 0.21 | 0.71 | 0.06 | 0.05 | 0.03 |
| CapsIE - 16 | **0.78** | - | - | **0.97** | - | - |
| CapsIE - 32 | 0.75 | - | - | **0.97** | - | - |
| CapsIE - 64 | 0.72 | - | - | **0.97** | - | - |
| Projector - 1st Intermediate Embedding | | | | | | |
| CapsIE - 16 | - | - | **0.78** | - | - | **0.97** |
| CapsIE - 32 | - | - | 0.77 | - | - | **0.97** |
| CapsIE - 64 | - | - | **0.78** | - | - | **0.97** |