# OpenReview forum: "Capsule Network Projectors are Equivariant and Invariant Learners"
_ICLR.cc/2025/Conference — ICLR 2025 Conference Withdrawn Submission_

### Official Review · Reviewer_L5wr · 2024-11-03

**Soundness:** 3
**Presentation:** 3
**Contribution:** 3
**Rating:** 5
**Confidence:** 4

**Summary:**

The paper proposes a self-supervised invariant equivariant approach based on Capsule Networks (Sabour et al., 2017). More specifically, the paper proposes to replace the project head, typically implemented by MLPs, by a capsule-based component based on SRCaps (Hahn et al. 2019). In doing so, the invariant (the activations) and equivariant (the pose matrices) aspects of the output of the CapsNet are considered  for learning the new representation.

The proposed method is evaluated on the 3DIEBench dataset, where it pushes the performance previously obtained by the state-of-the-art. Moreover, additional experiments on image in colour prediction tasks show the equivariant capabilities of the proposed method might go beyond than what it is trained for.

**Strengths:**

- The proposed method is sound. it is very clear how the characteristics of the inner-workings and representations of capsule networks match the invariance/equivariance goals targeted by the proposed method.

- The different design decisions made throughout the paper, in both the proposed method and conducted experiments, are clear and well motivated.

- The proposed method outperforms existing methods in the benchmark related to the considered 3DIEBench dataset.

- The paper has a good balance between verbal and formal presentation.

- As part of the validation of the proposed method, the paper includes an ablation study. This type of study is always helpful to assess the effect of the different parameters that define the proposed method.

**Weaknesses:**

- There is significant repetition between the contents of the Introduction (Sec.1) and Related Work (Sec.6) sections around the topics of equivariance and invariance. A revision/re-organization of this content may help improve the clarity/flow of the paper and help make additional room for extending the experimentation and/or discussion of already present experiments. Moreover, moving the description/introduction of Capsule Networks from Sec. 6.2. to an earlier section in the paper, would assist readers unfamiliar with this type of architecture.

- While the claim/goals of the paper regarding invariance and equivariance are general, the reported improvements are mostly constrained to experiments within the considered benchmark (orientation and location features from 3DIEBench). While the experiments on the colour prediction task (Sec. 5.2) are a good step towards the assessment of the general capabilities of the proposed method towards concurrently preserving invariance/equivariance, it is still too limited as to serve as grounds for generalization. The validation of the proposed method would strengthen by an extended evaluation considering other predictions problems (e.g. segmentation/localization of specific parts of the objects of interest, Skeleton-based pose estimation, in-painting, input reconstruction, etc. ) to assess the capabilities of the proposed method towards joint invariance/equivariance.

- While having an ablation study is desirable, it is currently missing an evaluation of the effect of the $\lambda$ weights applied to the loss functions that are part of the proposed method.

**Questions:**

- The results reported in Table.2 for the Color Prediction problem based on the Encoder Representation of SIE are significantly low. May you elaborate on why this is the case?

- May you indicate how the $\lambda$ weights for the loss functions introduced in Sec. 3.2 were selected?

---

### Official Review · Reviewer_HCE1 · 2024-11-04

**Soundness:** 2
**Presentation:** 3
**Contribution:** 2
**Rating:** 3
**Confidence:** 4

**Summary:**

This paper introduces CapsIE (Capsule Invariant Equivariant Network), a self-supervised architecture for learning both invariant and equivariant representations. CapsIE leverages Capsule Networks (CapsNets), an architecture that was introduced around 2017 and is considered somewhat outdated. The method incorporates a new entropy minimization objective function, claiming improved efficiency and downstream task performance with fewer parameters. Evaluations on the 3DIEBench dataset report state-of-the-art results, showing improvements over both previous self-supervised and supervised baselines. However, the experimental focus remains on a single dataset and task.

**Strengths:**

- Equivariant self-supervised learning is an interesting topic and is not systematically investigated in the literature.

- The entropy minimization objective introduced in the paper is novel. It is an interesting approach for enhancing the representation learning capacity of Capsule Networks.

- Promising initial results on the 3DIEBench dataset.

**Weaknesses:**

- My first main concern is that the proposed method relies heavily on Capsule Networks, which have been shown in prior research to be generally less robust and scalable compared to CNNs for a wide range of tasks (see [A]).

- My second main concern is the limited scope of experiments. The experiments are confined to a single dataset (3DIEBench). This narrow experimental scope significantly limits the impact of the results and raises concerns about the practical applicability of CapsIE.

- As mentioned in the paper, given the high computational cost associated with CapsNets, the practicality of deploying this method at scale remains questionable.

Based on these weaknesses (and limitations), I believe that the authors need to further develop the paper to make it suitable for publication in a top-tier conference like ICLR.

[A] Gu, Jindong, Volker Tresp, and Han Hu. "Capsule network is not more robust than convolutional network." Proceedings of the IEEE/CVF conference on computer vision and pattern recognition. 2021.

**Questions:**

See Weaknesses!

---

### Official Review · Reviewer_hGJH · 2024-11-10

**Soundness:** 3
**Presentation:** 3
**Contribution:** 2
**Rating:** 5
**Confidence:** 4

**Summary:**

The proposed CapsIE method leverages self-supervised CapsNets to learn equivariant and invariant representations, achieving state-of-the-art performance in downstream benchmarks. It shows significant improvements in self-supervised rotation and color prediction and outperforms supervised baselines. CapsIE also showcases the implicit strengths of its capsule architecture through experiments.

**Strengths:**

1. This is an intriguing work that empowers the inherent equivariance characteristics of capsule networks more effectively.
2. The significance of the proposed method has been demonstrated using more complex benchmarks compared to those traditionally used for validation in existing capsule network research.

**Weaknesses:**

1. The citation format is incorrect. Most citations should be within parentheses, but in all instances, they are presented without them.
2. While the methodology indeed enhances the equivariance of the original capsule network, there are still inherent limitations in the loss function that restrict its extension to real-world benchmarks. Real-world benchmarks often do not contain only a single (needed) equivariant factor; e.g., if there are five factors simultaneously, how would the loss function or the proposed self-supervised learning method handle such cases? Please discuss potential approaches for handling multiple equivariance factors simultaneously, or propose experiments to test the method's performance in such scenarios.
3. How should the current loss function be modified when trying to learn equivariance beyond those explicitly represented as scalars, such as color and rotation? Without prior human knowledge of answers, how can implicit yet useful equivariant factors present in benchmarks be learned? Please discuss potential modifications to the proposed approach for learning more complex equivariances, or suggest experiments to explore the method's ability to discover implicit equivariance factors without explicit supervision.

**Questions:**

Please address the concerns mentioned above in weaknesses.

---

### Note · Authors · 2024-11-24

**Comment:**

Due to a bug in our training code which effects results, we are withdrawing our paper. Thank you reviewers for your time.

**Withdrawal Confirmation:**

I have read and agree with the venue's withdrawal policy on behalf of myself and my co-authors.